# Caco-2/HT29-MTX co-cultured cells as a model for studying physiological properties and toxin-induced effects on intestinal cells

Pascal Hoffmann[ID][1]*, Marion Burmester[1], Marion Langeheine[2], Ralph Brehm[2], Michael T. Empl[3], Bettina Seeger[ID][3¤], Gerhard Breves[1]

**1** Institute for Physiology and Cell Biology, University of Veterinary Medicine Hannover, Hannover, Germany, **2** Institute for Anatomy, University of Veterinary Medicine Hannover, Hannover, Germany, **3** Institute for Food Toxicology, University of Veterinary Medicine, Hannover, Germany

¤ Current address: Institute for Food Quality and Food Safety, University of Veterinary Medicine, Hannover, Germany

* pascal.hoffmann@tiho-hannover.de

**Data Availability Statement:** All relevant data are within the manuscript and its Supporting information files.

## Abstract

Infectious gastrointestinal diseases are frequently caused by toxins secreted by pathogens which may impair physiological functions of the intestines, for instance by cholera toxin or by heat-labile enterotoxin. To obtain a functional model of the human intestinal epithelium for studying toxin-induced disease mechanisms, differentiated enterocyte-like Caco-2 cells were co-cultured with goblet cell-like HT29-MTX cells. These co-cultures formed a functional epithelial barrier, as characterized by a high electrical resistance and the presence of physiological intestinal properties such as glucose transport and chloride secretion which could be demonstrated electrophysiologically and by measuring protein expression. When the tissues were exposed to cholera toxin or heat-labile enterotoxin in the Ussing chamber, cholera toxin incubation resulted in an increase in short-circuit currents, indicating an increase in apical chloride secretion. This is in line with typical cholera toxin-induced secretory diarrhea in humans, while heat-labile enterotoxin only showed an increase in short-circuit-current in Caco-2 cells. This study characterizes for the first time the simultaneous measurement of physiological properties on a functional and structural level combined with the epithelial responses to bacterial toxins. In conclusion, using this model, physiological responses of the intestine to bacterial toxins can be investigated and characterized. Therefore, this model can serve as an alternative to the use of laboratory animals for characterizing pathophysiological mechanisms of enterotoxins at the intestinal level.

## Introduction

Infectious gastrointestinal diseases are often caused by pathogen-secreted toxins affecting the physiological function of the intestinal epithelium. One of those toxins is the cholera toxin (CTX), which is produced by *Vibrio cholerae* and which was, according to the World Health Organization, associated with 172,454 cholera cases as well as 1,304 deaths in 42 countries in

**Funding:** This project was funded by the Federal State of Lower Saxony in the joint project R2N – "Replace" and "Reduce" in Niedersachsen (Lower Saxony) – Alternative methods to replace or reduce animal models in biomedical research. No specific person was funded but the project itself. Additionally, this publication was supported by open access funds from Deutsche Forschungsgemeinschaft and University of Veterinary Medicine Hannover, Foundation. The funding is not specific to one of the authors but to the publication itself. The funders had no role in study design, data collection and analysis, decision to publish, or preparation of the manuscript.

**Competing interests:** The authors have declared that no competing interests exist.

2015 [1]. However, much higher numbers of unreported cases are estimated [2]. Another pathogen-secreted toxin is the heat-labile enterotoxin (HLTX), which is produced by *Escherichia coli*. Both toxins, CTX and HLTX, share the same well-characterized mode of action [3–5] and show a high degree of sequence homology [6–8] as well as approximately 80% structural similarity [9]. Both toxins bind to the monosialotetrahexosylganglioside (GM1) [10,11] and are taken up by cells via endocytosis [12]. After relocation via early endosomes [13], the Golgi apparatus and the endoplasmic reticulum [14], the toxin is released to the cytosol where it mediates the increase of intracellular cyclic adenosine monophosphate (cAMP) levels by activation of adenylate cyclase [15]. This leads to the phosphorylation of the cystic fibrosis transmembrane conductance regulator (CFTR) by protein kinase A (PKA) and therefore to a $Cl^-$ secretion [16,17]. The secretion of $Cl^-$ is further associated with an increasing $Na^+$ concentration in the intestinal lumen due to increasing paracellular $Na^+$ permeability [18], which ultimately results in severe diarrhea caused by the osmosis-driven high water flow induced by the hypertonic salt concentration in the intestinal lumen [19]. In many studies, mainly rodents have been used as laboratory animals to study these bacterial effects mechanistically. Within the framework of the 3Rs concept, the reduction of animal experiments is more desirable than ever, as there is growing evidence that species-specific *in vitro* models are more suitable to study molecular mechanisms than the classically used rodent models [20]. One way to avoid animal experiments is to use human-derived cell culture-based systems. Caco-2 cells, which are derived from a human colorectal carcinoma, represent a well-characterized cell culture system that displays enterocyte-like morphology and functionality, comparable to the small intestines, when differentiated by using certain culture conditions [21–26]. In addition to enterocytes, the second most abundant type of cells in the intestines are mucus-producing goblet cells [27]. These can be modelled using HT29-MTX cells, a sub-strain of the colorectal carcinoma-derived cell line HT29 and which can differentiate into goblet cell-like cells during cultivation [28]. The enterocyte-like Caco-2 cells have often been used to study bacterial infections [29,30] as well as toxin-induced effects on the epithelium [31,32], while HT29-MTX cells have mainly been used for studying bacterial adhesion and survival [33]. Both cell lines can be co-cultured to obtain a more physiological model of the small intestines [34–36], whereby the goblet cells provide a mucus layer that, as under *in vivo* conditions, serves as an important barrier against pathogens and toxins [reviewed by 37]. The small intestine itself is a typical target for CTX-producing *V. cholerae* or HLTX-producing *E. coli* [38–41]. The co-cultivation of Caco-2 and HT29-MTX cells was already published in 1996 [42]. Since then, studies of a broad field of scientific questions ranging from transport studies [43,44] to work on the adhesion of pathogens [45,46] and the association of nanoparticles [47] have been performed using this model. However, the measurement of functional and structural properties, followed by measurements on the response to bacterial toxins have never been performed using the Ussing chamber technique in combination with this cellular setup. The Ussing chamber enables the recording of transepithelial transport processes in the absence of electrochemical driving forces via the connection to a voltage clamp. This clamp generates the so-called short-circuit current ($I_{sc}$), which is directed against the current generated by electrogenic epithelial transport processes [48]. Therefore, the aim of the present study was firstly, to validate and to characterize physiological transport processes and barrier functions of co-cultured Caco-2 and HT29-MTX cells and secondly, to study the response of these cells to bacterial toxins, in order to investigate the suitability of this co-culture model for studying infectious diseases *in vitro* and to establish it as a potential alternative to classical toxicity studies using laboratory animals.

## Materials & methods

### Cell culture

Caco-2 cells (passages 10–18) were obtained from the German Collection of Microorganisms and Cell Cultures (DSMZ, Braunschweig, Germany) and HT29-MTX cells (passage 8–16) were obtained from the Max Rubner-Institut (MRI, Karlsruhe, Germany). Both cell lines were cultured in Dulbecco's Modified Eagle Medium (DMEM) containing 10% fetal bovine serum, 1% (v/v) non-essential amino acids, 2 mM L-glutamine, 100 μg/ml streptomycin and 100 U/ml penicillin (all purchased from Biochrom [Merck], Berlin, Germany). The cells were cultured for 28 days at 37˚C in a carbogen (95% $O_2$, 5% $CO_2$) atmosphere, with the culture medium being changed every two to three days. For Ussing chamber experiments and the Lucifer yellow assay, cells were cultured on Snapwells® (Corning, Kaiserslautern, Germany; diameter: 12 mm; pore size: 0.4 μm) containing 0.5 ml medium in the apical and 3.0 ml in the basolateral compartment., 1 x $10^5$ Caco-2 or HT29-MTX cells were seeded in each insert individually, while, for co-culture experiments, the cells were seeded in a 9:1 (Caco-2:HT29-MTX) ratio [36]. Mycoplasm test was performed every 2 weeks by PCR.

### TEER measurements

To determine cell monolayer integrity, transepithelial electrical resistance (TEER) was measured using an epithelial volt-ohm-meter ($EVOM^2$; WPI, Berlin, Germany). Measurements were performed at each culture medium exchange according to the manufacturer's instructions.

### Permeability study

In addition to TEER measurements, Lucifer yellow (100 μM, Sigma-Aldrich) dissolved in phenol red-free DMEM (Biochrom) was added to the apical compartment of the Snapwells® to determine the integrity of the cellular layer. This was performed as published before [49], with slight modifications. After an incubation period of 1, 3, 6 and 9 h, samples were taken from the apical (5 μl) and basolateral (15 μl) compartments and transferred to a black half-area 96-well plate (Greiner Bio-One, Kremsmünster, Austria), before the volume in each well was filled up to 100 μl with phenol red-free DMEM. Finally, Lucifer yellow quantification was performed by measuring fluorescence at an excitation wavelength of 425 nm and an emission wavelength of 530 nm.

### Ussing chamber experiments—Transport physiology

Cells cultured on Snapwells® were mounted in Ussing cambers [50], mimicking the mucosal and the serosal side of the intestine. Ussing chambers were connected to a computer-controlled voltage clamp (K. Mussler, Aachen, Germany). Each compartment was filled with 5 ml of the respective buffers (Table 1), which was heated to 37˚C and aerated with carbogen. After an equilibration phase of 5 min, the clamp was set to short-circuit conditions.

**Ussing chamber—Cellular characterization.**    Fifteen min after the application of short-circuit conditions, the cells were incubated with glucose (10 mM, mucosal, diluted in aqua destillata) and mannitol (10 mM, serosal, diluted in aqua destillata) for 20 min, followed by a 10 min incubation period with carbachol (10 μM, serosal, Sigma-Aldrich, diluted in aqua destillata) and a final incubation with forskolin (10 μM, serosal, Sigma-Aldrich, diluted in Dimethyl sulfoxide) for another 10 min. Indomethacin (10 μM) was added to impede prostaglandin synthesis and to avoid spontaneous $Cl^-$ secretion [51]. Throughout the experiment, both, the $I_{sc}$ and resistance ($R_t$) were measured. In each experiment, three chambers were run as replicates.

**Table 1. Composition of buffer solutions used for Ussing chamber experiments (if not stated otherwise, all chemicals were obtained from Merck, Darmstadt, Germany).** All chemicals were diluted in aqua destillata.

| | Mucosal buffer [mM] | Serosal buffer [mM] |
|---|---|---|
| NaCl | 119.6 | 119.6 |
| KCl | 5.4 | 5.4 |
| HCl | 0.4 | 0.4 |
| $MgCl_2 * 6 H_2O$ | 1.2 | 1.2 |
| $CaCl_2 * 2 H_2O$ | 1.2 | 1.2 |
| $NaHCO_3$ | 15.0 | 15.0 |
| $Na_2HPO_4$ | 1.2 | 1.2 |
| $NaH_2PO_4$ | 0.3 | 0.3 |
| Mannitol | 20.0 | 10.0 |
| Glucose | - | 10.0 |
| Indomethacin (Sigma-Aldrich, Schnelldorf, Germany) | 0.01 | 0.01 |

**Ussing chamber—Incubation with bacteria-derived toxins.** Fifteen min after the clamp was set to short-circuit conditions, the cells (Caco-2, HT29-MTX or co-cultures,) were incubated for 45 min with either 7.5 μg/ml CTX (CAS 9012-63-9; Enzo Life Sciences, Lörrach, Germany, diluted in aqua destillata) or 7.5 μg/ml HLTX (E8015, Sigma-Aldrich, diluted in aqua destillata). This was followed by the addition of 10 μM forskolin to the serosal chamber for 10 min and a final incubation of 20 min with 100 μM ouabain (Sigma-Aldrich, diluted in aqua destillata) on the serosal side. $I_{sc}$ and $R_t$ were measured continuously and replicates were obtained as mentioned above.

## Harvest of cells and preparation of apical membranes

Cells were harvested after cultivation on 10 cm petri dishes under the above-mentioned conditions. The culture medium was removed, 5 ml ice-cold phosphate buffered saline (PBS) was added, the cells were detached with a cell-scraper and the plate was rinsed twice with PBS to remove residual cells. Then, the cells were centrifuged at 3.000 x $g$ and 4˚C for 15 min. Afterwards, the supernatant was discarded and the pellet was either used to prepare the apical membrane (AM) or stored at -20˚C until further analysis. $CaCl_2$-precipitation was used to prepare the AM precipitation as previously described [52].

**SDS-polyacrylamide gel electrophoresis (SDS-PAGE) and immunoblotting.** The protein expression of sodium-dependent glucose co-transporter (SGLT1), its phosphorylated form pSGLT1, CFTR and the peptide transporter 1 (PepT1) was investigated by SDS-PAGE and immunoblotting. AM samples were denatured in Laemmli buffer containing 2% (w/v) SDS and 60 mmol/l dithiothreitol (DTT). 10 μl of each protein extract were loaded on an 8.5% (w/v) acrylamide gel. After gel electrophoresis (25 min at 60 V, followed by 80 min at 120 V), proteins were blotted (90 min at 90 V) onto a nitrocellulose membrane (VWR, Hannover, Germany). Then, after blocking for 90 min (see Table 2 for details), the membranes were incubated overnight at 4˚C with the primary antibody, followed by incubation with the secondary antibody. The detection of chemiluminescent signals was performed using a Chemidoc system (Bio-Rad Laboratories, München, Germany) and the band intensity was measured using the Image Lab software (version 6.1.0; Bio-Rad laboratories). After detection, the pSGLT1-membranes were stripped (0.2 M glycine, 0.05% [v/v] Tween-20, 1% [v/v] SDS, pH 2.0) and blocked again as described above. Incubation with an anti-SGLT1 antibody and corresponding secondary antibody as well as subsequent detection of chemiluminescence were performed as described above. Expression of villin was used as marker for AM and whole protein content as

**Table 2. Incubation conditions and dilution factors of the different antibodies used in the present study.**

|  | Denaturing conditions | Primary antibody dilution factor and blocking buffer* | Secondary antibody dilution factor† |
|---|---|---|---|
| PEPT1[1] | 95˚C, 5 min | 1:500; 5% BSA/TBST | 1:2,000[8] |
| SGLT1[2] | 40˚C, 15 min | 1:2,000; 5% M TBST | 1:20,000[7] |
| pSGLT1 Ser[418] [3] | 40˚C, 15 min | 1:200; 5% M TBST | 1:15,000[7] |
| CFTR[4] | 95˚C, 5 min | 1:1,000; 5% M TBST | 1:2,000[6] |
| Villin[5] | 95˚C, 5 min | 1:5,000; 5% TBST | 1:25,000[9] |

* Incubated at 4˚C overnight;

† Incubated at room temperature for 90 min.

[1] monoclonal mouse-anti-PEPT1 (373742, Santa Cruz, lot no.: 31916).

[2] polyclonal rabbit- anti-SGLT1 (ab14686, Abcam, UK, lot no.: JBC17887303).

[3] monoclonal rabbit-antipSGLT1, custom made by Perbio Science, Germany (epitope: KIRKRApSEKELMI).

[4] monoclonal rabbit-anti-CFTR (#78335, Cell Signaling Technology, lot no.: 1).

[5] monoclonal mouse-anti-Villin (1D2C3, Santa Cruz, lot no.: L0808).

[6] goat-anti-rabbit IgG HRP-conjugated (#7074, Cell Signaling Technology, lot no.: 26).

[7] goat-anti-rabbit IgG HRP-conjugated (A9169, Sigma-Aldrich).

[8] mouse-IgGκ BP-HRP-conjugated (516102, Santa Cruz).

[9] goat-anti-mouse IgG HRP-conjugated (A2304, Sigma-Aldrich).

BSA: Bovine serum albumin; TBST: Tris-buffered saline including 0.1% Tween 20; M: Skimmed dry milk.

housekeeping protein. Further information on antibodies and incubation conditions is given in Table 2.

## Histological analysis

**Analysis of mucus layer formation.** Mucus layer formation was analyzed by means of mucin staining. Cells were cultivated on Snapwells® as described above. Then, the Snapwell® membranes were cut out, fixed in Bouin solution for 10 min and rinsed three times with PBS. After fixation, the membranes were cut in two pieces and embedded in 5% (w/v) agarose followed by embedding in paraffin. For morphological evaluation, 3 μm-slices were sectioned and hematoxylin and eosin (H & E) staining was performed. In order to confirm the existence of goblet cells, de-paraffinized slides were stained with periodic acid-Schiff reagent (PAS), dehydrated and mounted with Eukitt® (O. Kindler GmbH, Freiburg, Germany), all according to standard protocols.

**Expression of zonula occludens-1.** Immunohistochemical staining of zonula occludens-1 protein (ZO-1) was obtained by fixation and embedding of cells on Snapwell® membranes as described in above. After de-paraffinization and inhibition of endogenous peroxidase activity with 3% $H_2O_2$ in 80% ethanol for 30 min, sections were microwave-pretreated with sodium citrate buffer (pH 6.0) for 3 x 5 min at 800 Watts. The slides were then allowed to cool down to room temperature for 30 min and blocked with 3% BSA for 20 min, before they were incubated with the primary anti-ZO-1 antibody (Santa Cruz Biotechnology, Dallas, USA; catalog no.: sc-10804, dilution factor: 1:80) overnight at 4˚C. The sections were then exposed to the secondary antibody (biotin-labelled goat-anti-rabbit; Vector, Burlingame, USA; catalog no.: BA-1000; dilution factor: 1:200) for 60 min at room temperature and, after rinsing, 30 min with the ABC-System (Vector; catalog no.: PK-6100). After visualization with 3,3′-Diaminobenzidine (DAB), the sections were counterstained with hematoxylin for 10 sec and rinsed with running tap water. Finally, the slides were dehydrated and mounted with Eukitt®. Isotype controls were included in the analyses and images were captured using a Zeiss Axioskop

(Zeiss, Jena, Germany) mounted with an Olympus SC50 camera controlled by the Olympus CellSens software (Olympus Soft Imaging Solutions GmbH, Germany). Numeric apertures (NA) of the objective lenses used where: 10 x/NA 0.3; 20 x/NA 0.5 and 40 x/NA 0.75.

Immunofluorescence of ZO-1 was performed by cultivation of Caco-2, HT29-MTX and co-cultivation of HT29-MTX and Caco-2 cells on glass covers for 7 and 28 days, with a medium change occurring every 2 or 3 days. Then, fixation was performed by replacing the medium with 3% paraformaldehyde for 10 min at room temperature, followed by three times rinsing with PBS. After fixation, cells were permeabilized for 10 min using 0.25% Triton X-100/PBS and blocked with 5% goat-serum in PBST for an hour. The primary anti-ZO-1 antibody was diluted 1:100 in goat-serum and incubated with the slides at 4˚C overnight. After washing thrice with PBS, incubation with the secondary fluorescence-labelled antibody (goat anti-rab-bit) was performed for 60 min at room temperature. Cover glasses were embedded using Pro Long Gold (Invitrogen, Carlsbad, USA). No staining of the nuclei was performed to minimize eventual background staining. Analysis was performed using a Keyence BZ-X800 microscope and the BZ-X800 analyzer software (Keyence GmbH, Hannover, Germany). Numeric aper-tures of the objective lense used was: 60x/NA 1.4.

## Data analysis and statistics

Basal values of $I_{sc}$ and $R_t$ were determined by calculating the arithmetic mean of the last 10 data points before the addition of an agent. Changes in short-circuit currents ($\Delta I_{sc}$) were calcu-lated by subtracting the arithmetic mean from the maximal value after the addition of each agent. Densitometrical analysis of protein expression was performed by normalizing the signal of specific protein bands to the total protein content. The normal distribution of the residuals was tested using the D'Agostino & Pearson test. A paired *t*-test was used to compare datasets containing only two groups. When comparing datasets containing three or more groups, the following statistical tests were used: a one-way ANOVA followed by Tukey's post-hoc test was used for normally distributed data, while the Kruskal-Wallis test followed by Dunn's multiple comparisons post-hoc test was applied to non-normally distributed datasets. All statistical analyses were performed using Prism version 8.0.1 (GraphPad, San Diego, USA) and *p* values $\leq$ 0.05 were considered statistically significant.

## Results

### Cellular monolayer integrity increases during cultivation time

The integrity of the cellular monolayer was determined at each medium change until the cells were used for experimental purposes. After four weeks of cultivation, Caco-2 (555 ± 52.9 Ω * cm$^2$) and co-cultured Caco-2/HT29-MTX cells (495.7 ± 57.8 Ω * cm$^2$) reached significantly higher TEER-values when compared with HT29-MTX cells (291.3 ± 52.8 Ω * cm$^2$; Fig 1). After 9 hours of incubation with Lucifer yellow, no specific fluorescence signal could be detected in the basolateral chamber (data not shown). Expression of the tight junction associated protein ZO-1 was examined by immunohistochemical and immunofluorescence staining. Immuno-histochemical staining of ZO-1 at day 28 appears to be increased on the cell surface in relation to the signal to day 7. The isotype control was negative (Fig 2).

Immunofluorescence staining of ZO-1 in Caco-2 cells showed a positive signal at the cellu-lar boards. This appears in lines for cells cultivated for one week, Caco-2 cells cultivated for 28 days showed a more diffuse staining at the intracellular connections but less staining in the cytoplasm. Cultivation of HT29-MTX cells for 7 days showed a ZO-1 signal in the cytoplasm, while cells cultivated for 28 days showed a ZO-1 signal at the cell-cell connections but not in the cytoplasm. Co-cultivated Caco-2 and HT29-MTX cells showed a ZO-1 signal at the cell-

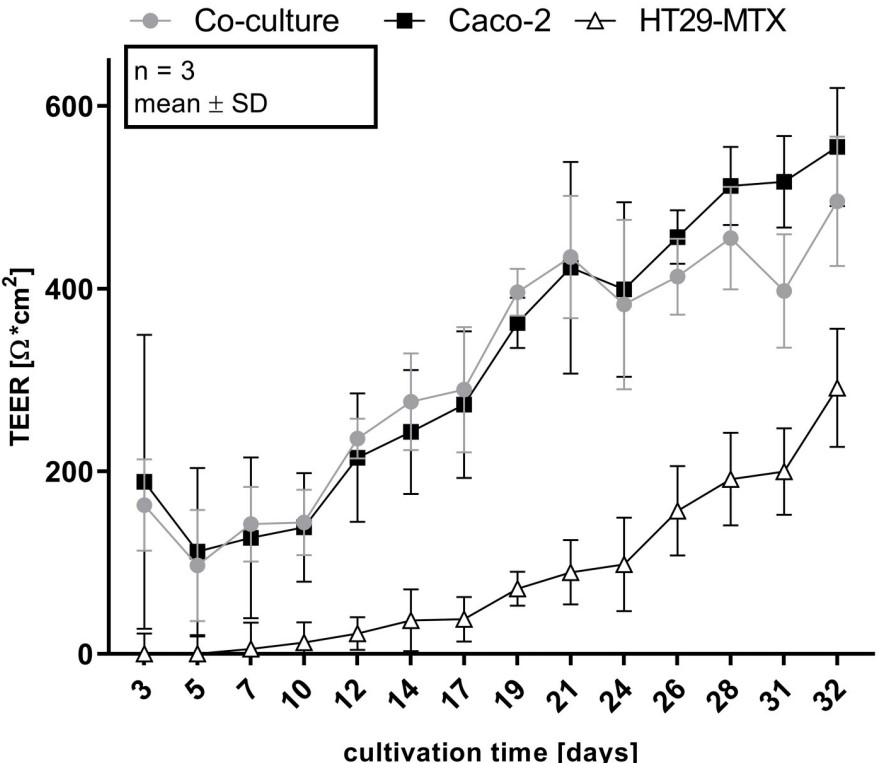

**Fig 1. Changes of TEER values as a function of time of Caco-2, HT29-MTX and co-cultured Caco-2 and HT29-MTX cells incubated for 32 days on Snapwells®.** Shown are values obtained from four experiments, each performed in triplicates.

cell connections and the cytoplasm when cultivated for 7 days, which was enhanced in cells co-cultivated for 28 days. The negative control showed only a low background in all stainings (Fig 3).

## Mucus layer formation

Caco-2, HT29-MTX and co-cultures of both cell lines grown on Snapwell® membranes were stained with H & E as well as hematoxylin and PAS to determine cellular arrangement and the formation of mucus. All cell types showed a similar growth pattern and cells with a larger lumen and more basal oriented nuclei can be detected in the co-culture (green triangles, Fig 4). While hematoxylin and PAS staining showed no specific signal in Caco-2 cells, a strong signal in all HT29-MTX cells as well as some cells in the co-culture can be seen (violet triangles, Fig 4). Interestingly, some areas in the co-culture show a PAS-signal on top of PAS-negative cells (black triangles, Fig 4).

## Ussing chamber studies—Transport characteristics

Basal $R_t$ values were significantly lower in HT29-MTX cells ($227.1 \pm 51.83 \; \Omega * cm^2$) when compared to Caco-2 cells ($462.5 \pm 65.84 \; \Omega * cm^2$) and co-cultured cells ($387.1 \pm 117.6 \; \Omega * cm^2$), while basal $I_{sc}$ values were not different between the different cell line incubations. The mucosal addition of glucose resulted in a significant increase in $I_{sc}$ in co-cultured and Caco-2 cells, while HT29-MTX did not respond to this treatment (Fig 5). No increase in $I_{sc}$ was induced by the addition of carbachol to Caco-2 cells, while $I_{sc}$ significantly increased in co-

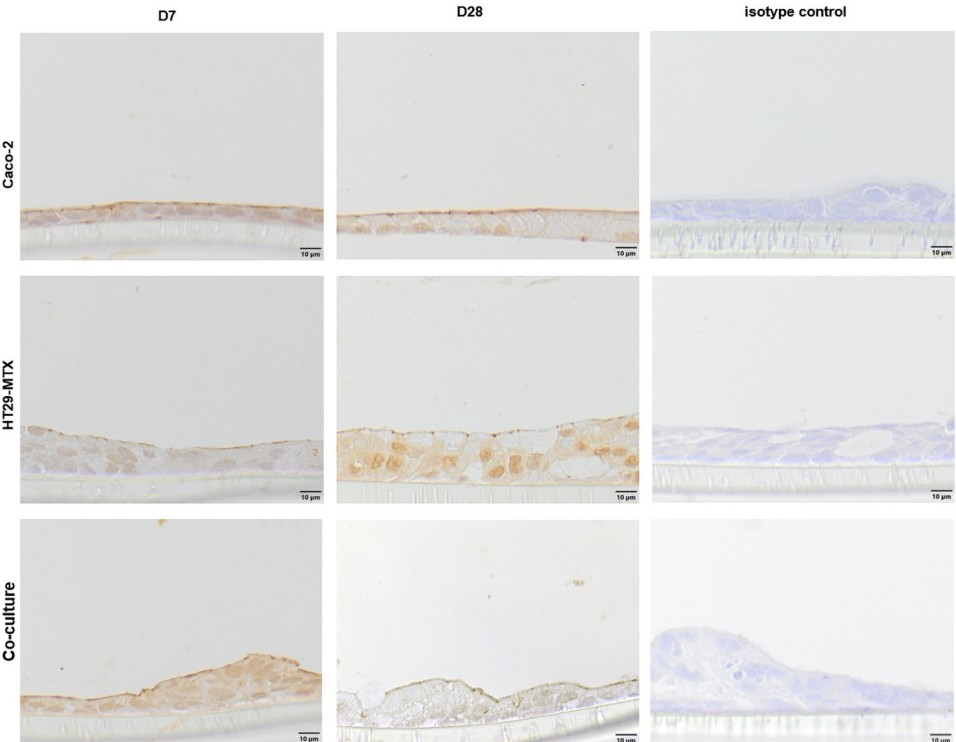

**Fig 2. Immunohistochemical staining (including isotype controls) of ZO-1 (cross-section) in Caco-2, HT29-MTX as well as co-cultured Caco-2 and HT29-MTX cells cultivated for 7 (D7) and 28 days (D28).** Representative figures are shown, chosen from three independent experiments with three technical replicates per cellular approach.

cultured cells and HT29-MTX cells upon exposure to this agent. Forskolin induced a significant $I_{sc}$ increase in all three cell types.

## Ussing chamber studies—Effects of bacterial toxins

Mucosal incubation with CTX led to a significant increase in $I_{sc}$ in all cellular cultures (Fig 6). After treatment with CTX, chloride-secretion was stimulated by forskolin, resulting in a significantly higher $I_{sc}$ in HT29-MTX cells. Ouabain led to a significant decrease in $I_{sc}$ in Caco-2, and HT29-MTX co-cultures.

Mucosal addition of HLTX led to a significant increase in $I_{sc}$ in Caco-2 cells, while HT29-MTX and co-cultured cells did not respond to this treatment, while subsequent addition of forskolin induced an effect in all cellular incubations (Fig 7). In contrast, ouabain led to a decrease in the $I_{sc}$ in Caco-2- and HT29-MTX cells as well as in the co-cultured HT29-MTX and Caco-2 cells.

## Structural characterization of protein expression reveals different abundance of transport proteins

Preparations of Caco-2 and co-cultures of Caco-2 and HT29-MTX cells showed a band at $\approx$ 90 kDa when incubated with antibodies against SGLT1 and pSGLT1, while expression of this glucose transporter is absent in HT29-MTX cells (Fig 8). In case of SGLT1, this was further confirmed by densitometry while pSGLT1 expression showed no significant differences (Fig 9). Villin was detected as a $\approx$ 90 kDa band in all preparations, albeit with a less intense signal

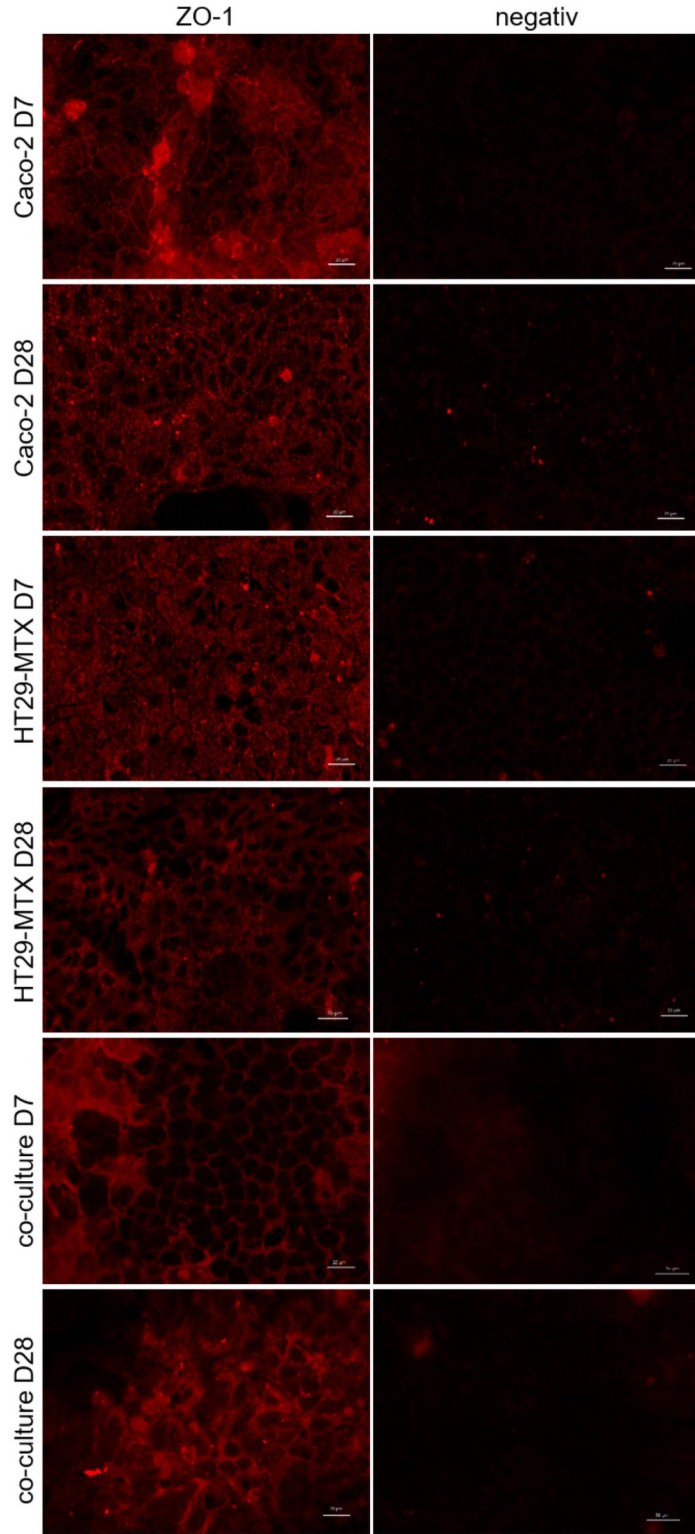

**Fig 3. Immunofluorescence staining (top view) of ZO-1 in Caco-2, HT29-MTX and co-cultured Caco-2 and HT29-MTX cells cultivated for 7 (D7) and 28 days (D28).** ZO-1 is displayed in red. Representative figures are shown, chosen from three independent experiments with three technical replicates per cellular approach.

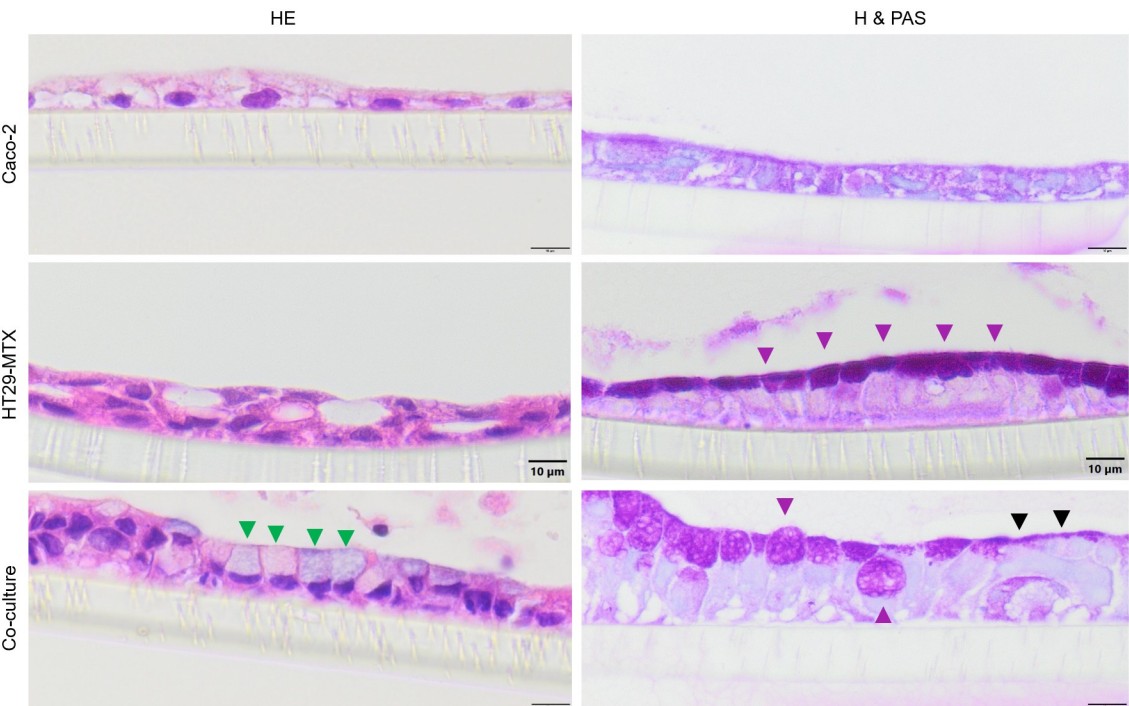

**Fig 4. Cellular arrangement (cross sections) of cells grown on Snapwells® for 28 days.** Cells were either stained with hematoxylin and eosin (H & E) or hematoxylin and PAS (H & PAS). Green triangles indicate cells with a large lumen, violet triangles indicate PAS-positive cells and black triangles indicate PAS-positive staining on top of PAS-negative cells. Representative figures are shown, chosen from three independent experiments with three technical replicates per cellular approach.

in HT29-MTX cells, which was confirmed in the frame of the densitometric analysis (Fig 9). A PepT1 band at $\approx$ 70 kDa was detected in all cell types. A similar result was obtained in the case of CFTR, which showed a band at $\approx$ 189 kDA.

## Discussion

### Layer integrity is characterized by increasing electrical resistance as well as low permeability due to elevated tight junction protein expression

The TEER values of $\approx$ 500 $\Omega * cm^2$ for co-cultured and Caco-2 cells and $\approx$ 300 $\Omega * cm^2$ for HT29-MTX cells are in accordance with previously published results and suggest that the cells

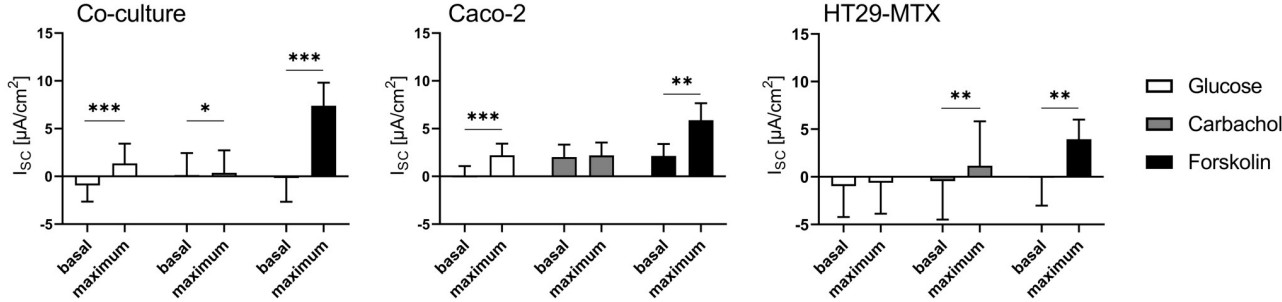

**Fig 5. Basal and maximal $I_{sc}$ values in Caco-2, HT29-MTX and co-cultured Caco-2 and HT29-MTX cells determined before and after subsequent addition of glucose, carbachol and forskolin.** Values shown are the mean ± SD of four independent experiments, each performed in triplicates. * $p < 0.05$; ** $p < 0.01$; *** $p < 0.001$.

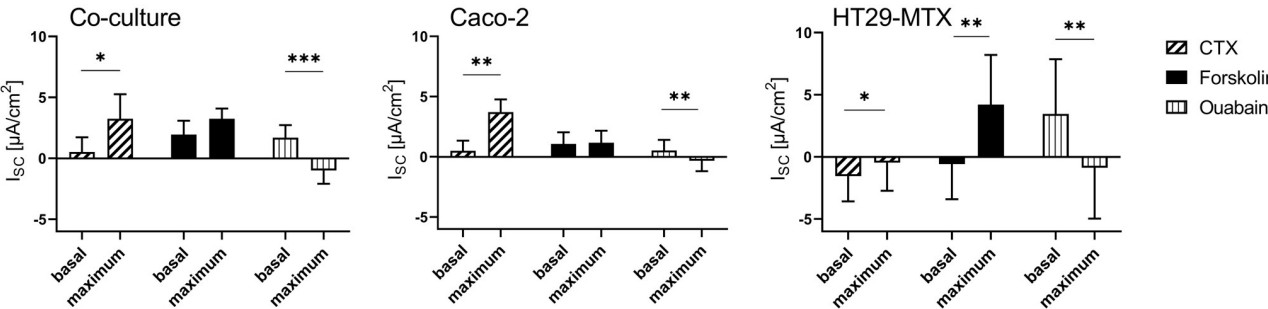

**Fig 6. Basal and maximal $I_{sc}$ values in Caco-2, HT29-MTX and co-cultured Caco-2 and HT29-MTX cells determined before and after subsequent addition of CTX, forskolin and ouabain.** Values shown are the mean ± SD of four independent experiments, each performed in triplicates. $^{*}$ $p < 0.05$; $^{**}$ $p < 0.01$; $^{***}$ $p < 0.001$.

form an intact monolayer [35,53,54]. Absolute TEER values obtained in the Ussing chamber are generally lower than when measured using the volt-ohm-meter due to different methodical setups, although relative resistance values between the two systems and cell types are comparable. In the Lucifer yellow assay, no paracellular transport was detected after 9 hours of incubation indicating the expression of tight junction proteins [35], a finding which is supported by the increased expression of the tight junction associated protein ZO-1 [27,34,42] which is associated with increasing TEER values indicating tight junction expression [55–57] and is commonly used as marker for barrier integrity [58]. ZO-1 is expressed in all cellular compositions after 7 days but the expression is changed after 28 days cultivation time. However, the modulation and change of ZO-1 expression has been shown [59] and is responsible for the assembly of tight junction proteins such as claudin [60]. It was discussed that the reason for unspecific staining in the co-cultured cells at D28 is due to accumulation of the antibody in the produced mucus [61] which would also be the case for the mucus producing HT29-MTX cells. Cellular organization is comparable to existing literature [62] and shows mucus producing goblet cells, mucus covered areas as well as cells which do not produce mucus.

## Cellular characterization shows functional transport characteristics

Glucose transport is mediated by SGLT1 [63], which results in an increase in $I_{sc}$, as shown in co-cultured and Caco-2 cells but not in HT29-MTX cells. This can be confirmed by the expression of SGLT1 in Caco-2 cells [21], while HT29-MTX, being of goblet cell lineage [28], do not

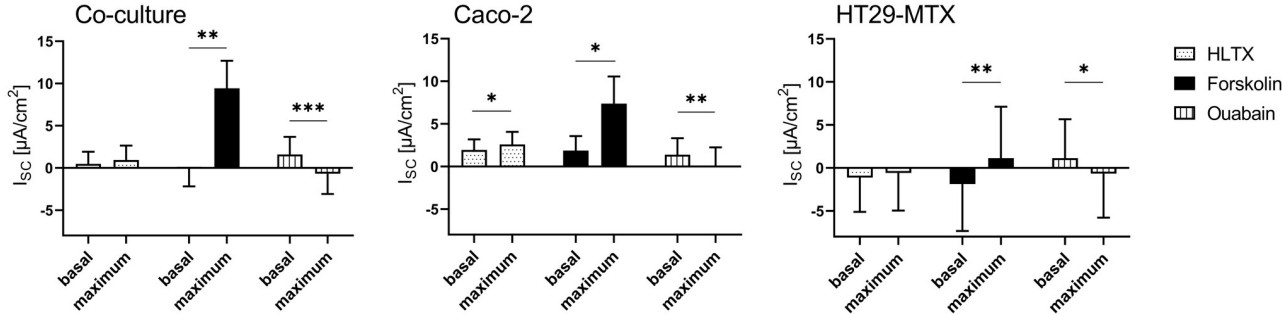

**Fig 7. Basal and maximal $I_{sc}$ values in Caco-2, HT29-MTX and co-cultured Caco-2 and HT29-MTX cells determined before and after subsequent addition of HLTX, forskolin and ouabain.** Values shown are the mean ± SD of four independent experiments, each performed in triplicates. $^{*}$ $p < 0.05$; $^{**}$ $p < 0.01$; $^{***}$ $p < 0.001$.

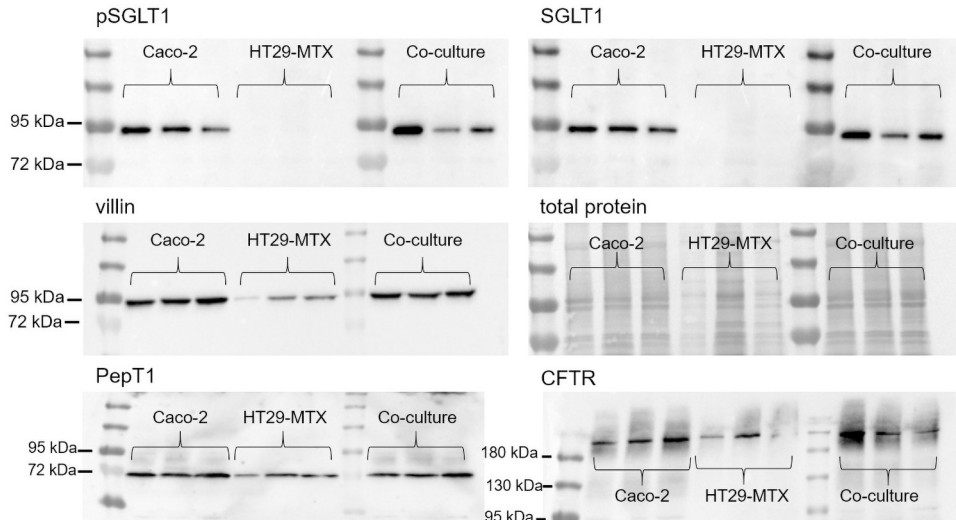

**Fig 8. Protein expression of pSGLT1, SGLT1, villin, total protein content, PepT1 as well as CFTR in Caco-2, HT29-MTX and co-cultured Caco-2 and HT29-MTX cells.** pSGLT1 and SGLT1 were blotted on the same membrane, which was stripped after incubation with the pSGLT1-antibody. Each lane on each membrane represents a biological replicate, experiments were performed in triplicates.

express this transporter in the present as well as other studies [64], Villin was enriched in the AM preparations, with weaker expression in HT29-MTX cells. Glucose transport in co-cultures of Caco-2 and HT29-MTX cells is most likely mediated by the relatively high amount of SGLT1-expressing Caco-2 cells in this incubation.

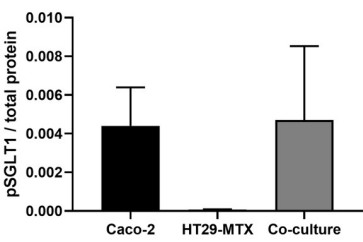
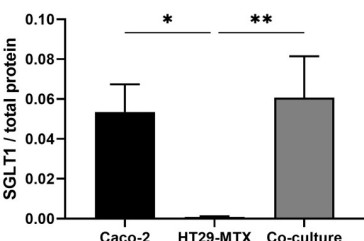
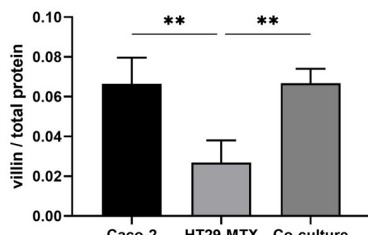

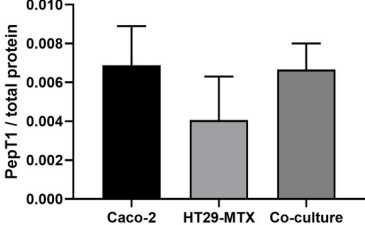
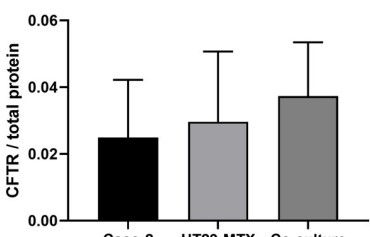

**Fig 9. Densitometrical analysis of Western blots performed with pSGLT1, SLT1, villin, PepT1 and CFTR in relation to total protein content in Caco-2, HT29-MTX and co-cultured Caco-2 and HT29-MTX cells.** Values shown are the mean ± SD of three independent experiments. * p < 0.05; ** p < 0.01.

Carbachol induces a Cl$^-$ secretion in a Ca$^{2+}$-dependent manner [65], which results in an increase in the $I_{sc}$ by stimulating calcium activated calcium channels (CaCC) [66]. The comparatively high response of HT29-MTX cells to this compound could be explained by the ability of these cells to produce high amounts of mucus, which is concomitant to an elevated chloride secretion [67]. In contrast, Caco-2 cells are not able to produce such high amounts of mucus [68] and therefore release lower amounts of Cl$^-$. Nonetheless, even at higher concentrations of carbachol, the changes in $I_{sc}$ of Caco-2 cells appear to remain small [69].

Forskolin activates adenylate cyclase, which results in an intracellular increase of cyclic adenosine monophosphate [70], which in turn has been described to lead to a Cl$^-$-secretion via CFTR followed by an increase in $I_{sc}$ in Caco-2 [71] and HT29-MTX cells [72]. However, the higher response of co-cultured cells to forskolin can be explained by a higher activation of CFTR. The expression of PepT1 as a transporter for di- and tripeptides [reviewed by 73] was observed in all cellular incubations, as has previously been shown in Caco-2 cells [74,75] and HT29-MTX cells [34].

In conclusion, the existence of epithelial transport systems in the apical membrane confirms the functional integrity of the cell cultures as a valid model for the small intestinal epithelium.

## Toxin incubation leads to pathophysiological responses

Cholera toxin derived from *V. cholera* causes secretory diarrhea by activating adenylate cyclase [15], which leads to an increase in intracellular cAMP levels and results in a decrease in sodium absorption and an increase in chloride secretion [76]. In human colonic epithelial T84 cells, an increase in cAMP-stimulated chloride secretion has been shown to be mediated either by phosphorylation of CFTR [17] or by recruitment of the transporter to the AM [77]. Similar results obtained in Caco-2 cells in the present study as well as in previous publications after an incubation period of approximately 30 min [78]. An influence of CTX on mucus release in intact mucosa has also been shown [79], albeit not in HT29 cells [80]. However, an effect of CTX on the $I_{sc}$ was observed in HT29-MTX cells in the present study. Thus, it may be concluded that this cellular subtype shows, a behavior more comparable to the actual *in vivo* situation regarding its response to CTX. In addition, co-cultured cells showed a similar response to CTX as Caco-2 cells, thus supporting the assumption that this co-culture model has the potential to mimic the human small intestinal epithelium in a physiological way [34,35,81]. An increase in $I_{sc}$ after treatment with CTX impaired the response to forskolin, although an additional activation of CFTR by forskolin can be neglected due to the continuous effect of CTX and the ensuing permanent activation of the adenylate cyclase [82]. HLTX and CTX are about 80% similar regarding their structure [83]. However, CTX showed an effect in Caco-2, HT29-MTX and co-cultured Caco-2/HT29-MTX cells in the Ussing chamber, whereas HLTX only affected Caco-2 cells and was less pronounced. This is comparable to the *in vivo* situation where infections with HLTX-producing *E. coli* are milder and of shorter duration when compared to *V. cholerae* infections as reviewed by Spangler [3]. In spite of this structural similarity, other factors such as the tissue colonization capacity of the pathogens themselves in the *in vivo* situation or the different A2 domain expressed in these two toxins may explain this discrepancy [4,84]. The A2 domain is responsible for the linking between the A1 domain, which is responsible for the adenylate cyclase activity, and the pentameric B subunit which binds to G$_{M1}$-ganglioside in the target cell [9,85,86].

In this study, fetal bovine serum is used for cultivation of the cells according to established culture conditions of the cell lines and to ensure comparability to earlier studys [21,34,35]. However, it is desirable to replace this serum by an animal-free alternative while the

comparability to the serum-using system must be verified and compared with other studies already using serum-free medium [87].

The present study characterizes a model of the human small intestine regarding several physiological (e.g. glucose transport) and pathophysiological reactions to bacterial toxins (e.g. chloride secretion) using the Ussing chamber technology. The combination of Caco-2 and HT29-MTX cells has proven to be an improvement over the widely used classical Caco-2 model of cellular transport due to the presence of a cell-covering mucus layer. Moreover, glucose transport and tight junction expression correlate with functional endpoints such as the $I_{sc}$ or TEER measured in the Ussing chamber and other systems used herein. In conclusion, this study serves as proof of concept showing that the Caco-2/HT29-MTX co-culture model is a suitable system for studying various physiological intestinal functions ranging from nutrient transport to reactions to pathogens or toxins without using laboratory animals or material derived thereof. Possible future endpoints that could be analyzed using this system include studying the influence of bacterial toxins on nutrient transport, layer integrity or their interactions with pharmaceutical agents.

## Supporting information

**S1 File.**
(XLSX)

**S1 Raw images. Raw images of Western Blot membranes of pSGLT1, SGLT1, villin, total protein content, PepT1 and CFTR.**
(PDF)

## Acknowledgments

The authors wish to thank Prof. Pablo Steinberg for providing the HT29-MTX cells.

## Author Contributions

**Conceptualization:** Pascal Hoffmann, Gerhard Breves.

**Data curation:** Pascal Hoffmann.

**Formal analysis:** Ralph Brehm.

**Funding acquisition:** Gerhard Breves.

**Investigation:** Pascal Hoffmann, Marion Burmester, Marion Langeheine, Ralph Brehm, Michael T. Empl.

**Methodology:** Pascal Hoffmann, Marion Burmester, Marion Langeheine, Ralph Brehm, Michael T. Empl, Bettina Seeger.

**Project administration:** Gerhard Breves.

**Resources:** Gerhard Breves.

**Supervision:** Gerhard Breves.

**Validation:** Gerhard Breves.

**Visualization:** Ralph Brehm.

**Writing – original draft:** Pascal Hoffmann, Gerhard Breves.

**Writing – review & editing:** Ralph Brehm, Michael T. Empl, Bettina Seeger, Gerhard Breves.

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
