## [Decision Letter · Decision Letter 0]

27 Apr 2021

PONE-D-21-06276

Caco-2/HT29-MTX co-cultured cells as a model for studying physiological properties and toxin-induced effects on intestinal cells

PLOS ONE

Dear Dr. Hoffmann,

Thank you for submitting your manuscript to PLOS ONE. After careful consideration, we feel that it has merit but does not fully meet PLOS ONE’s publication criteria as it currently stands. Therefore, we invite you to submit a revised version of the manuscript that addresses all the points raised during the review process.

Two experts have evaluated the manuscript and found it of interest for the field, but needs amendments at several parts. There might be some technical problems with the permeability measurements (extensive dilution of the samples?) which need to be repeated for the meaningful characterization of the model. The same is true for the junctional stainings. 

We look forward to receiving your revised manuscript.

Kind regards,

Mária A. Deli, M.D., Ph.D.

Academic Editor

PLOS ONE

Journal Requirements:

Reviewers' comments:

Reviewer's Responses to Questions

**Comments to the Author**

1. Is the manuscript technically sound, and do the data support the conclusions?

Reviewer #1: Partly

Reviewer #2: Yes

2. Has the statistical analysis been performed appropriately and rigorously? 

Reviewer #1: Yes

Reviewer #2: Yes

3. Have the authors made all data underlying the findings in their manuscript fully available?

Reviewer #1: Yes

Reviewer #2: Yes

4. Is the manuscript presented in an intelligible fashion and written in standard English?

Reviewer #1: Yes

Reviewer #2: Yes

5. Review Comments to the Author

Reviewer #1: In this manuscript Hoffmann and colleagues described and characterized a co-culture model of the human intestinal Caco-2 epithelial and the mucus producing HT29-MTX goblet cell lines which is a good cell combination for investigating the molecular mechanisms of various enterotoxins. The cholera toxin and heat-labile enterotoxin show structural similarity and cause electrophysiological changes (Cl- outflow) in the human intestinal barrier that leads to diarrhea. The effect of these toxins was tested on their barrier model and the electrophysiological changes were characterized by sort-circuit current. As a novelty, they used the Ussing chamber technique to characterize this intestinal model, in which the cell layers form mechanical and functional barriers. Using this system, the electrophysiological and structural changes can be measured simultaneously. Although the present study is of interest for the field, there are several points which need to be clarified, especially the characterization of co-co-culture model using Ussing chamber and better explanation of the short-circuit current technique.

Comments:

1. For the introduction:

I suggest the authors to add more detail about the background of the Ussing chamber and the “short-circuit current” technique, why was this method chosen, what are the advantages and its importance. Please also describe the mechanism of the investigated toxins. A justification is also needed for the selection of the studied transporters.

In addition, since the message of the present study is to highlight the connection between toxin effect and the cellular electrophysiological changes by measuring short-circuit current condition, a short description on the effect of toxins and ion transporters would be needed for a better understanding of the significance of the research for the larger public.

2. The Caco-2 epithelial cell line forms a tight barrier resulting in low paracellular permeability for several marker molecules. Lucifer yellow (LY) was used in the present study as a paracellular marker to assess the tightness of the barrier model. The molecular weight of LY is small (444 Da) and the incubation time was quite long (9 hours), so it is hard to understand the result that “no paracellular transport was detected”. Based on the literature an apparent permeability coefficient in the range of 10-6 to 10-7 cm/s should be obtained on a similar model. I suggest to repeat this experiment with a better fluorescence spectroscopy detection technique. It is also suggested to use higher sample volumes from the basolateral chamber (full volume at the end of the experiment) without any further dilution. For the full characterization of a new model it is important to give the exact parameters of the paracellular permeability (eg. apparent permeability coefficient).

3. Immunofluorescence staining was performed for ZO-1 to prove the presence of tight junctions (TJs) between cells. While ZO-1, a junctional associated protein and a cytoplasmic linker between transmembrane TJ proteins and actin cytoskeleton, is accepted to indicate the presence of intercellular junctions, staining for integral membrane TJ proteins occludin or claudins would be better indicators of the presence of tight interepithelial junctions. TJ staining should be performed on monocultures, not just on co-culture to complete the TEER and immunohistochemical data.

Minor suggestions:

1. In line 99 a half bracket and in line 103 an empty bracket should be removed.

2. The word “briefly” occurs 4 times in the methods section. This word should be omitted, since this section must be as detailed as possible in a research article. The journal policy also supports full description of methods for reproducibility.

3. Figure 2: please replace the image of D28 for co-culture. It does not represent a layer. Also, the membrane is not visible in contrast to all other images.

4. Figure 8: a densitometry is needed to evaluate the WB changes.

Reviewer #2: I would like to note to the authors some differences in nomenclature, spelling, and notation:

In 43- line: “unreported cases are estimated (2) Another” the punctuation mark at the end of the sentence is missing.

In 103- line: “DMEM () was added” the origin of the substance is missing in bracket.

In 121- line: “aqua destillata” if you use the official latin name, or trade name then Aqua destillata or in English distilled water.

In 274- line and in “Ussing chamber studies” result part the authors use Ω * cm2 notation, but previously and in other chapters used Ω x cm2.

In 284-line authors use adenylyl cyclase name, previously in 356 line used adenylate cyclase. It would be obvious to use only one synonym because it can be misleading.

6. PLOS authors have the option to publish the peer review history of their article (what does this mean?). If published, this will include your full peer review and any attached files.

Reviewer #1: **Yes: **Dr. Alexandra Bocsik

Reviewer #2: No

---

## [Author Response · Author response to Decision Letter 0]

10 Aug 2021

The authors thank the reviewers for their time and efforts to review this manuscript giving us the opportunity to further enhance the quality of this study by adding useful comments.

Reviewer #1: In this manuscript Hoffmann and colleagues described and characterized a co-culture model of the human intestinal Caco-2 epithelial and the mucus producing HT29-MTX goblet cell lines which is a good cell combination for investigating the molecular mechanisms of various enterotoxins. The cholera toxin and heat-labile enterotoxin show structural similarity and cause electrophysiological changes (Cl- outflow) in the human intestinal barrier that leads to diarrhea. The effect of these toxins was tested on their barrier model and the electrophysiological changes were characterized by sort-circuit current. As a novelty, they used the Ussing chamber technique to characterize this intestinal model, in which the cell layers form mechanical and functional barriers. Using this system, the electrophysiological and structural changes can be measured simultaneously. Although the present study is of interest for the field, there are several points which need to be clarified, especially the characterization of co-co-culture model using Ussing chamber and better explanation of the short-circuit current technique.

Comments:

1. For the introduction:

I suggest the authors to add more detail about the background of the Ussing chamber and the “short-circuit current” technique, why was this method chosen, what are the advantages and its importance. Please also describe the mechanism of the investigated toxins. A justification is also needed for the selection of the studied transporters.

In addition, since the message of the present study is to highlight the connection between toxin effect and the cellular electrophysiological changes by measuring short-circuit current condition, a short description on the effect of toxins and ion transporters would be needed for a better understanding of the significance of the research for the larger public. 

The characteristic effects of the enterotoxins and their influence on the CFTR have been described in more detail in the introduction (lines 48-60), thus justifying the analysis of this transporter, in addition to an overview of the Ussing chamber methodology being added to the introductory section (lines 87-92). Finally, the focus on the SGLT1 and PepT1-transporter as characteristic examples for epithelial transport proteins of the small intestines was specified in the discussion (lines 411-413).

2. The Caco-2 epithelial cell line forms a tight barrier resulting in low paracellular permeability for several marker molecules. Lucifer yellow (LY) was used in the present study as a paracellular marker to assess the tightness of the barrier model. The molecular weight of LY is small (444 Da) and the incubation time was quite long (9 hours), so it is hard to understand the result that “no paracellular transport was detected”. Based on the literature an apparent permeability coefficient in the range of 10-6 to 10-7 cm/s should be obtained on a similar model. I suggest to repeat this experiment with a better fluorescence spectroscopy detection technique. It is also suggested to use higher sample volumes from the basolateral chamber (full volume at the end of the experiment) without any further dilution. For the full characterization of a new model it is important to give the exact parameters of the paracellular permeability (eg. apparent permeability coefficient).

The cells for this experimental approach were cultivated on a membrane with 0,4 µm pore size, which impairs the transport capacity more when compared to the more frequently used membranes with a 3 µm pore size . The high TEER values obtained in the present study imply a high cell-to-cell connection, which results in decreased paracellular transport and thus lower Papp values. In addition, we intentionally opted for a cultivation of cells on membranes with 3 µm pores, in order to be able to compare the results with those obtained in the Ussing chamber experiments. Generally, the non-detectable transport of Lucifer Yellow (LY) after an incubation period of 9 hours shows the high integrity of the cellular layer, resulting in transport processes observed in the Ussing chamber setup being attributable to trans- rather than paracellular transport mechanisms. In order to show that the LY transport assay was correctly performed, a representative evaluation of one LY experiment is attached for review, in which it is shown that all measured values are within the linear range of the assay. In addition, as indicated in the manuscript, the assay was performed in accordance with the following publication: Willenberg, Michael et al. (2015). In conclusion, the non-detectable paracellular transport of LY is intentional and desirable in the frame of the present study, as this allows the analysis of transcellular transport processes only. 

3. Immunofluorescence staining was performed for ZO-1 to prove the presence of tight junctions (TJs) between cells. While ZO-1, a junctional associated protein and a cytoplasmic linker between transmembrane TJ proteins and actin cytoskeleton, is accepted to indicate the presence of intercellular junctions, staining for integral membrane TJ proteins occludin or claudins would be better indicators of the presence of tight interepithelial junctions. TJ staining should be performed on monocultures, not just on co-culture to complete the TEER and immunohistochemical data. 

To give a broader insight in the expression of ZO-1, stainings of Caco-2 and Ht29-MTX monocultures were added to the manuscript and the Figure 3 was changed accordingly (lines 232-238; 267-275; 375-381). To keep the focus on the functional aspects of this study only ZO-1 staining as tight-junction associated protein was performed as it has been shown that ZO-1 directly correlates with higher expression of tight junction proteins and increasing TEER values (Dolan, Naughton et al. 2012, Chang, Liang et al. 2020, Reale, Huguet et al. 2021).

Minor suggestions:

1. In line 99 a half bracket and in line 103 an empty bracket should be removed. 

The text was corrected accordingly.

2. The word “briefly” occurs 4 times in the methods section. This word should be omitted, since this section must be as detailed as possible in a research article. The journal policy also supports full description of methods for reproducibility.

The word has been omitted and the full description of the used method is given.

3. Figure 2: please replace the image of D28 for co-culture. It does not represent a layer. Also, the membrane is not visible in contrast to all other images. 

The respective image has been replaced and a new figure (Figure 8) was created.

4. Figure 8: a densitometry is needed to evaluate the WB changes. 

A densitometrical analysis was performed (lines 181-182; 243-245; 343-360) and the respective figure added as Figure 9 to the manuscript. Furthermore, the materials & methods section was updated with information on densitometrical and statistical analysis of the protein expression. Results of the analysis were incorporated in the results and the discussion sections.

Reviewer #2: I would like to note to the authors some differences in nomenclature, spelling, and notation:

In 43- line: “unreported cases are estimated (2) Another” the punctuation mark at the end of the sentence is missing.

In 103- line: “DMEM () was added” the origin of the substance is missing in bracket.

In 121- line: “aqua destillata” if you use the official latin name, or trade name then Aqua destillata or in English distilled water.

In 274- line and in “Ussing chamber studies” result part the authors use Ω * cm2 notation, but previously and in other chapters used Ω x cm2.

In 284-line authors use adenylyl cyclase name, previously in 356 line used adenylate cyclase. It would be obvious to use only one synonym because it can be misleading.

All the above-mentioned comments regarding nomenclature, spelling and notation have been taken into account and changed accordingly in the manuscript.

Chang, Y.-N., Y. Liang, S. Xia, X. Bai, J. Zhang, J. Kong, K. Chen, J. Li and G. Xing (2020). "The High Permeability of Nanocarriers Crossing the Enterocyte Layer by Regulation of the Surface Zonal Pattern." Molecules (Basel, Switzerland) 25(4): 919.

Dolan, B., J. Naughton, N. Tegtmeyer, F. E. B. May and M. Clyne (2012). "The Interaction of Helicobacter pylori with the Adherent Mucus Gel Layer Secreted by Polarized HT29-MTX-E12 Cells." PLOS ONE 7(10): e47300.

Reale, O., A. Huguet and V. Fessard (2021). "Co-culture model of Caco-2/HT29-MTX cells: A promising tool for investigation of phycotoxins toxicity on the intestinal barrier." Chemosphere 273: 128497.

Willenberg, I., M. Michael, J. Wonik, L. C. Bartel, M. T. Empl and N. H. Schebb (2015). "Investigation of the absorption of resveratrol oligomers in the Caco-2 cellular model of intestinal absorption." Food Chem 167: 245-250.

---

## [Decision Letter · Decision Letter 1]

13 Sep 2021

Caco-2/HT29-MTX co-cultured cells as a model for studying physiological properties and toxin-induced effects on intestinal cells

PONE-D-21-06276R1

Dear Dr. Hoffmann,

We’re pleased to inform you that your manuscript has been judged scientifically suitable for publication and will be formally accepted for publication once it meets all outstanding technical requirements.

Kind regards,

Mária A. Deli, M.D., Ph.D.

Academic Editor

PLOS ONE

Additional Editor Comments (optional):

I trust the authors to correct the figure legend, as requested by the reviewer, during the final proofreading of their manuscript.

Reviewers' comments:

Reviewer's Responses to Questions

**Comments to the Author**

1. If the authors have adequately addressed your comments raised in a previous round of review and you feel that this manuscript is now acceptable for publication, you may indicate that here to bypass the “Comments to the Author” section, enter your conflict of interest statement in the “Confidential to Editor” section, and submit your "Accept" recommendation.

Reviewer #1: (No Response)

2. Is the manuscript technically sound, and do the data support the conclusions?

Reviewer #1: Yes

3. Has the statistical analysis been performed appropriately and rigorously? 

Reviewer #1: Yes

4. Have the authors made all data underlying the findings in their manuscript fully available?

Reviewer #1: Yes

5. Is the manuscript presented in an intelligible fashion and written in standard English?

Reviewer #1: Yes

6. Review Comments to the Author

Reviewer #1: (No Response)

7. PLOS authors have the option to publish the peer review history of their article (what does this mean?). If published, this will include your full peer review and any attached files.

Reviewer #1: No

---

## [Editor Report · Acceptance letter]

29 Sep 2021

PONE-D-21-06276R1 

Caco-2/HT29-MTX co-cultured cells as a model for studying physiological properties and toxin-induced effects on intestinal cells 

Dear Dr. Hoffmann:

I'm pleased to inform you that your manuscript has been deemed suitable for publication in PLOS ONE. Congratulations! Your manuscript is now with our production department. 

Kind regards, 

on behalf of

Dr. Mária A. Deli 

Academic Editor

PLOS ONE